# Noncoding RNAs Are Promising Therapeutic Targets for Diabetic Retinopathy: An Updated Review (2017–2022)

**DOI:** 10.3390/biom12121774

**Published:** 2022-11-28

**Authors:** Mengchen Wang, Qiaoyu Li, Meiqi Jin, Zhen Wang, Xuelian Zhang, Xiaobo Sun, Yun Luo

**Affiliations:** 1Institute of Medicinal Plant Development, Peking Union Medical College and Chinese Academy of Medical Sciences, Beijing 100193, China; 2Beijing Key Laboratory of Innovative Drug Discovery of Traditional Chinese Medicine (Natural Medicine) and Translational Medicine, Beijing 100193, China; 3Key Laboratory of Bioactive Substances and Resource Utilization of Chinese Herbal Medicine, Ministry of Education, Beijing 100193, China; 4NMPA Key Laboratory for Research and Evaluation of Pharmacovigilance, Beijing 100193, China

**Keywords:** diabetic retinopathy, miRNA, circRNA, lncRNA

## Abstract

Diabetic retinopathy (DR) is the most common complication of diabetes. It is also the main cause of blindness caused by multicellular damage involving retinal endothelial cells, ganglial cells, and pigment epithelial cells in adults worldwide. Currently available drugs for DR do not meet the clinical needs; thus, new therapeutic targets are warranted. Noncoding RNAs (ncRNAs), a new type of biomarkers, have attracted increased attention in recent years owing to their crucial role in the occurrence and development of DR. NcRNAs mainly include microRNAs, long noncoding RNAs, and circular RNAs, all of which regulate gene and protein expression, as well as multiple biological processes in DR. NcRNAs, can regulate the damage caused by various retinal cells; abnormal changes in the aqueous humor, exosomes, blood, tears, and the formation of new blood vessels. This study reviews the different sources of the three ncRNAs—microRNAs, long noncoding RNAs, and circular RNAs—involved in the pathogenesis of DR and the related drug development progress. Overall, this review improves our understanding of the role of ncRNAs in various retinal cells and offers therapeutic directions and targets for DR treatment.

## 1. Introduction

Diabetic retinopathy (DR), one of the most frequent microvascular complications of diabetes, has become the leading cause of adult blindness worldwide. DR clinically manifests as retinal vascular abnormalities that facilitate DR diagnosis. DR mainly occurs in two stages: non-proliferative diabetic retinopathy (NPDR) and proliferative diabetic retinopathy (PDR) [1]. NPDR is the early stage of DR in which hemorrhages and microaneurysms can be detected using fundus photography; patients are usually asymptomatic at this stage. PDR is an advanced stage of DR, characterized by neovascularization, and patients may experience vitreous hemorrhage or traction retinal detachment that can lead to vision loss [2]. Diabetic macular edema (DME) is the main cause of acquired vision loss in patients with DR and may occur in the early or late stages of DR [3]. The retina comprises 55 different cell types, including pigment epithelium cells, photoreceptor cells, bipolar cells, ganglion cells, horizontal cells, amacrine cells, and Müller cells (Figure 1), each of which has its own functions [4]. The pathological findings of DR include aberrant changes such as inflammation, excessive cell proliferation, and apoptosis in pericytes, endothelial cells (ECs), ganglial cells, retinal pigment epithelium cells (RPEs), and glial cells [5]. 

To date, controlling blood glucose, blood pressure, and blood lipids has been recognized as the most effective way to slow the progression of DR. Long-term exposure to hyperglycemia, hypertension, and hyperlipidemia lead to microangiopathy and eventually to DR. Timely laser treatment can slow vision loss in PDR and DME [6]. Except for vitrectomy, the intraocular injection of anti-vascular endothelial growth factor (anti-VEGF) drugs, and the injection of steroids, there exists no effective drug for DR treatment [7]. Moreover, a clinical trial has shown that the use of VEGF drugs for DR treatment is not effective in all patients and that it may even cause safety issues [8]. Therefore, novel therapeutic targets are warranted for developing new treatments for DR.

In recent years, noncoding RNAs (ncRNAs), including micro-RNAs (miRNAs), circular RNAs (circRNAs), and long noncoding RNAs (lncRNAs), have been implicated in regulating gene expression and affecting various biological processes in DR [9]. ncRNAs have been found in cells, exosomes, extracellular vesicles (EVs), and blood in recent years. Thus, in this review, we reviewed the research progress of noncoding RNAs in the prevention and treatment of DR in recent five years, and evaluate the potential pathological mechanisms of DR from the aspects of isolated sources and potential therapeutic drugs. We also review the clinical or preclinical therapeutic drugs, including traditional Chinese medicine and Western medicine, and emphasize a few potential therapeutic drugs, which can provide a promising direction for exploring therapeutic drugs.

## 2. miRNAs

miRNAs are small noncoding RNA molecules with a length of 20–24 nucleotides that play a role in regulating gene expression in essential biological processes in animals and plants [10]. An miRNA can act on multiple target genes, and several miRNAs in combination can finely regulate the expression of the same genes [11]. Some miRNAs have been shown to be involved in several important processes in life activities, including early development, cell proliferation, cell differentiation, cell apoptosis, cell death, and fat metabolism [12]. miRNAs perform their functions by destabilizing their mRNA targets to regulate the translation of proteins involved in many diabetic complications such as retinopathy, nephropathy, and wound healing [13]. For example, miR-21 promotes DR pathogenesis by downregulating PPARα [14]. We discuss below the recent research progress on miRNAs in terms of various DR-related cells, exosomes, EVs, blood, and aqueous humor (AH) in this review. Figure 2 summarizes the research progress on the pathogenesis of DR caused by miRNA and its target molecules.

### 2.1. ECs

Retinal ECs line the microvasculature and supply retinal nerves with abundant oxygen and other nutrients. These cells also protect the retina’s blood–retinal barrier by examining the retina for underlying inflammation or infection [15]. The presence of several miRNAs that play a crucial role in the DR process, has been reported in EC injury stimulated by high glucose (HG) levels. miR-590-3p reduces pyroptotic death by targeting NLRP1 in an HG-induced EC injury model [16]. EC proliferation and migration are suppressed by miR-126 in rats with streptozotocin (STZ)-induced DR [17]. In retinal tissue, miR-301a-3p stimulates the expression of inflammatory factors and apoptosis, as observed in a DR rat model [18]. miR-124-3p expression decreases in HG-stimulated ECs, whereas miR-124-3p overexpression suppresses HG-induced EC injury [19]. miR-15b inhibits vascular abnormalities in rats with DR by reducing the formation of endothelial tubes and VEGF expression [20].

miR-431-5p is expressed at higher levels in serum extracellular vesicles and ECs of patients with DR compared with those of healthy controls; hence, miR-431-5p may be a novel biomarker for early diagnosis and precise monitoring [21]. miR-135b-5p induces HIF1α expression in DR mice, which promotes EC proliferation and angiogenesis [22]. Moreover, miR-29a downregulation can lead to basement membrane thickening, which contributes to the pathogenesis of DR [23] and miR-139-5p promotes retinal neovascularization by facilitating tube formation and increasing VEGF protein levels in ECs [24]. miR-423-5p regulates the viability of retinal endothelial cells and promotes neovascularization by targeting HIPK2 [25]. In addition, it can also directly target TFF1 to regulate HG-induced apoptosis of RPE cells [26].

### 2.2. Pericytes

Pericytes are branched cells present throughout the body and are particularly abundant in retinal microvessels. These cells play an important role in the formation and stability of blood vessels, promote the formation of the blood–brain and blood–retinal barriers, and regulate communication with the ECs. Reduction in the number of pericytes is the focal point of DR pathogenesis [27]. MiR-138-5p in pericytes protects from early DR by regulating and promoting cell proliferation [28]. miR-199a-3p and miR-15b are expressed in pericytes at low levels, which promotes DR by accelerating cell proliferation and migration [29,30]. 

### 2.3. Retinal Ganglion Cells

The output of retinal information in the vertebrate visual system is transmitted by several dozens of retinal ganglion cells, and damage to these ganglion cells is the main cause of visual impairment [31]. Bipolar and amacrine cells transmit visual information from photoreceptors to retinal ganglions, which, in turn, transmit the information to the neurovisual system of the brain [32]. The downregulation of miR-495 appears to alleviate HG-induced apoptosis of RGCs through Notch1/PTEN/Akt signaling [33]. Moreover, high glucose levels induce retinal ganglion cell injury by increasing the expression levels of miR-145-5p [34]. 

### 2.4. Glial Cells

Human retinal glial cells include three types: Müller cells, microglia, and astrocytes [35]. Müller cells are the main glial cells in the retina, involved in the construction of the blood–retinal barrier [36]; they maintain the retinal integrity and are involved in regulating the supply of nutrients to various parts of the eye. Disruption of Müller cell function results in the formation of new blood vessels and retinal neuronal dysfunction [37]. The expression level of miR-365 generally increases and promotes Müller cell gliosis by targeting Timp3 [38]. A recent report showed that Müller glia, which are the sole source of exosomal miR-9-3p, can be transferred to retinal ECs, which may then activate VEGFR2 phosphorylation to promote angiogenesis in DR [39]. A report showed that miR-125a-5p prevented vascular leakage induced by macrophages by targeting Ninjurin1 [40]. miR-29a/b could ameliorate DR by impairing Müller cell function via Forkhead box protein O4 [41].

### 2.5. RPE Cells

RPE cells are an important component of the eye. Some studies have shown that RPE is involved in the development of DR [42]. RPE cells can be found between the neural retina and the choriocapillaris. RPE cells perform the following functions in the eye: blood–retinal barrier formation, providing nutritional support, oxidative damage prevention, ionic homeostasis maintenance, production and secretion of growth factors, and maintenance of ocular immune privilege, among others [43]. miR-195 promotes cell permeability in HG-stimulated RPEs [44]. miR-5195-3p protects RPE cells from high glucose-induced harm, which could be a potential therapeutic target for DR [45]. miR-126 counteracts the proangiogenic effects of hyperglycemia in RPE cells [46]. miR-139-5p significantly decreases ARPE-19 cell apoptosis, oxidative stress, and inflammation caused by HG levels [47]. Moreover, miR-144 can protect RPE cell injury from oxidative stress, a prominent inducer of DR [48]. TIMP3 is the direct target of miR-122, which can promote apoptosis of high glucose-induced human retinal pigment epithelial cells by regulating TIMP3 [49]. Moreover, miR-122 is significantly increased in the serum of DR patients, suggesting that miR-122 can be used as a biomarker to determine the progression of DR [50].

### 2.6. Blood and Tears

The initial stage of DR is asymptomatic, and currently available treatments for DR mainly focus on the advanced stage of the disease, i.e., when microvascular complications appear and cause irreversible structural changes. The treatment method in the early stage is mainly focused on strictly controlling DR risk factors, such as blood glucose, blood pressure, and blood lipids [51]. Therefore, novel methods to detect the initial stage of DR are warranted. Serum and ocular biomarkers, which are capable of assessing inflammation and microvascular formation, have been widely used in the early diagnosis and prediction of various diseases and are useful in detecting the progression of DR [52]. miR-20a-5p, miR-20a-3p, miR-20b, miR-106a-5p, miR-27a-5p, miR-27b-3p, miR-206-3p, and miR-381-3p are dysregulated in the retina and serum of mice with DR and could, therefore, be potential pharmacological targets for DR [53]. A clinical study that performed experimental research suggested that serum miR-211 acts as a biomarker for DR by modulating sirtuin 1 [54].

Moreover, plasma miR-26a-5p levels and retinal nerve fiber layer thickness are positively correlated; this may be a potential biomarker for early-stage DR [55]. Sun et al. found that miRNA-23a levels are significantly decreased in the blood and tears of patients with DR. Overexpression of miRNA-23a significantly inhibits retinal microvascular EC proliferation by reducing VEGF expression [56]. miR-1281 was found to be considerably overexpressed in serum samples from patients with DR and could, therefore, be a new target in treating DR [57].

### 2.7. EVs

As a result of their size and origin, EVs can be divided into three main categories: exosomes, microvesicles, and apoptotic bodies. EVs are found in the blood, urine, cerebrospinal fluid, saliva, and other body fluids and can promote intercellular communication by transporting functionally active biological molecules, including various miRNAs, and induce physiopathological changes [58]. miR-192 derived from mesenchymal stem cell-secreted EVs can delay the inflammatory response and angiogenesis in DR [59]. EVs do not just contain miRNAs; they also deliver miRNAs to cells. EVs secreted from mesenchymal stem cells inhibit apoptosis and reduce inflammatory responses by targeting miRNA-18b in ECs [60]. Although the biology of EVs in DR is not well investigated, its important role in DR has been recognized [61]. 

Exosomes are extracellular vesicles with a diameter of 30–150 nm and have a powerful capacity for intercellular communication, neovascularization, repair regeneration, and immune regulation. Exosomes can be secreted by almost all human cells; they play a major role in cardiovascular diseases, cancers, and other conditions [62]. Exosomes are now being increasingly studied in the context of eye diseases. Moreover, exosomes can inhibit retinal atrophy and relieve retinal ischemia [63]. Compared with that in healthy individuals, miR-25-3p and miR-320b are present in higher levels, whereas miR-495-3p is present in lower levels in plasma exosomes extracted from patients with DR [64]. Exosomal miR-107 derived from ECs was found to ameliorate the profibrotic phenotypes of pericytes by directly targeting the HIF-1α/Notch1 axis, thereby benefitting the treatment of pulmonary fibrosis [65]. 

miR-486-3p induced by bone marrow mesenchymal stem cell-derived exosomes play a protective role in DR by repressing the TLR4/NF-κB axis [66]. Coincidentally, exosomal miRNA-17-3p derived from human umbilical cord mesenchymal stem cells ameliorates inflammatory reactions and antioxidant injury in mice with DR [67]. Exosomes also mediate cell-to-cell crosstalk, and miR-202-5p derived from RPE exosomes suppresses endothelial-to-mesenchymal transition in DR [68].

### 2.8. AH

AH is produced by ciliary bodies and is a colorless and transparent tissue fluid. It provides nutrition to the iris, cornea, and lens and also maintains intraocular pressure [69]. AH has been reported as an emerging resource of miRNA in DR. One study reported on miRNA profiling of AH from patients with diabetic macular edema [70]. Consequently, another study reported several identified miRNAs in AH, which may support the pathogenesis of diabetic macular edema [71]. We hypothesized that miRNAs derived from AH may serve as new biomarkers or targets for DR.

## 3. circRNA

circRNAs are a unique class of endogenous noncoding RNAs that are widely expressed in mammalian serum and are more abundant, have a higher specificity, and high organization than other RNA types [72]. circRNA can regulate the expression and inhibit the transcription of miRNA; this characteristic is the reason circRNAs are part of various pathways in the human body; circRNAs bind protein factors and play a key role in the growth and development of organisms [73]. Recently, circRNAs were demonstrated to serve as a sponge for miRNAs and proteins as well as to regulate gene expression, epigenetic modifications, peptide translation, and pseudogene formation [74]. circRNAs have been identified and characterized in patients with DR [75] and have accordingly been regarded as novel therapeutic targets for DR [76]. A number of studies have demonstrated that some circRNAs play an integral role in the development of DR, thus affecting the proliferation and migration of retinal ECs as well as the formation of endothelial angiogenesis [75]. We discuss below the recent research progress on circRNAs in various DR-related cells, blood, and vitreous humor (VH). Figure 3 summarizes the main circRNAs involved in DR pathophysiology.

### 3.1. ECs

The expression of circZNF609 in ECs significantly increases with high glucose and hypoxia stress levels, and the knockdown of circZNF609 prevents retinal vessel loss and suppresses pathological angiogenesis [77]. circ_001209 deteriorates diabetic retinal vascular function by regulating miR-15b-5p and COL12A1 [78]. circCOL1A2 enhances miR-29b expression to promote proliferation, migration, angiogenesis, and vascular permeability in EC [79]. In HG conditions, circ_0001879 promotes the proliferation and migration of human HRMECs [80]. circ_0002570 suppresses the proliferation, migration, and angiogenesis of hRMECs based on the miR-1243/angiomotin axis [81]. Overexpression of Circ_0005015 regulates miR-519d-3p; exerts inhibitory effects on STAT3, MMP-2, and XIAP; and regulates retinal EC function [75]. Downregulation of CircDMNT3B regulates miR-20b-5p and BAMBI to promote the formation of diabetic retinal blood vessels, thereby providing a potential therapeutic strategy for DR [82].

### 3.2. Pericytes

Pericytes are key regulators of vascular development, stabilization, and permeability in DR [83]. Recently, cZNF532 overexpression was found to ameliorate diabetes-induced pericyte degeneration by increasing the expression of NG2, LOXL2, and CDK2 [84]. cPWWP2A, which is transported by exosomes in pericytes, inhibits miR-579 activity, leading to increased expression of angiopoietin 1, SIRT1, and occludin, subsequently alleviating diabetes mellitus-induced retinal vascular dysfunction [85].

### 3.3. RPEs

circ_0084043 induces RPE cell injury by triggering the Wnt/β-catenin signal pathway in response to high glucose [86] and miR-140-3p-mediated TGFA gene expression [87]. circ-ITCH inhibits the expression of MMP-2, MMP-9, and TNF-α in DR by reducing the expression of miR-22 [88]. circ_0041795 promotes RPE cell injury by sponging miR-646, directly targeting VEGFC [89]. circZNF532 downregulation protects RPEs against HG-induced apoptosis and pyroptosis, [90] whereas circPSEN1 inhibition ameliorates ferroptosis of HG-treated RPEs [91].

### 3.4. Blood and Tears

Li et al. determined the expression profiles of serum exosomal mRNAs, miRNAs, and circRNAs and verified that CircFndc3b plays a significant role in the angiogenesis of DR [92]. Wu reported that hsa_circ_0001953 may be a potential biomarker for DR in human blood [93].

### 3.5. VH

VH is a body fluid like blood. It is a colorless, extremely hydrated, gel-like substance that fills the spaces between the lens and retina of the eye. VH accounts for four-fifths of the entire volume of the eyeball and is the largest component of the eye [94]. Distinct structural changes in VH can be observed during the course of DR, and these changes reflect the pathological processes occurring in the tissues at the interface of the vitreous and retina [95], which opens up new opportunities for exploring various phenomena in DR. A recent study profiled circRNAs in VH and found 122 upregulated and nine downregulated circRNAs in the eyes of patients with DR versus healthy patients [96]. circFTO binds miR-128-3p, promotes diabetes-induced angiogenesis, and disrupts the blood–retinal barrier. TXNIP is a target gene of miR-128-3p and is expressed at high levels in DR models [97]. Thus, circRNAs in the VH may be a promising biomarker for DR.

## 4. IncRNAs

lncRNAs are noncoding RNAs of more than 200 nucleotides [98] and play important roles in many processes, including metabolism [99], viral infections [100], epigenetic regulation, cell cycle regulation [101], and cell differentiation regulation [102]. Accumulating evidence has suggested that IncRNAs regulate miRNAs in the progression of DR as well as the expression of related genes during transcription or epigenetic mechanisms to affect the progression of DR. Thus, lncRNAs are considered potential biomarkers for DR [103]. Wang et al. evaluated the effects of anti-VEGF drug treatment on lncRNA and mRNA expression in the fibrovascular membranes of patients with PDR. Based on their microarray results, 263 lncRNAs were upregulated and 164 were downregulated among 427 differentially expressed lncRNAs. Gene ontology and Kyoto Encyclopedia of Genes and Genomes analyses revealed that differentially expressed lncRNAs are involved in multiple PDR-related pathways and metabolic processes [104]. Figure 4 depicts the schematic overview of lncRNAs promoting the pathogenesis of DR.

### 4.1. ECs

lncRNA SNHG16 attenuates oxidative stress-induced pathological angiogenesis in ECs by regulating miR-195, which is a promising target for DR therapy [105]. Moreover, lncRNA MIR497HG inhibits the proliferation and migration of HRECs by targeting the miRNA-128-3p/SIRT1 axis [106]. lncRNA H19 targets miR-200b to indirectly regulate TGF-β1 signaling protein expression, thereby preventing endothelial–mesenchymal transition (EndMT) in DR [107]. lncRNA MEG3 also suppresses EndMT in DR models by inhibiting the PI3K/Akt/mTOR signaling pathway [108]. lncRNA VEAL2 can regulate PRKCB activity in zebrafish and human retinal vascular endothelium and regulate the endothelial permeability of DR [109]. lncRNA MALAT1 upregulates the expression of PDE6G via miR-378a-3p, promotes the proliferation of retinal vascular ECs, and inhibits apoptosis under HG conditions [110]. lncRNA MEG3 levels are significantly decreased in patients with DR, while serum VEGF and TGF-β1 levels are significantly increased. Therefore, the progression of DR can be inhibited by the overexpression of lncRNA MEG3 [111]. Knockdown of lncRNA RNCR3 can inhibit retinal EC proliferation and migration via the action of RNCR3/KLF2/miR-185-5p and alleviate retinal vascular dysfunction [112].

### 4.2. Pericytes

lncRNA MIAT regulates the pyroptosis of primary human retinal pericytes stimulated by advanced glycation end product-modified bovine serum albumin [113]. The expression of lncRNA–MALAT1 was significantly upregulated in the retina of diabetic rats and diabetic mouse models, and MALAT1 knockdown improved DR and reversed the severe pericyte loss caused by diabetes [114].

### 4.3. Gangliocytes

LncRNA–sox2OT accelerated the damage to retinal ganglion cells (RGCs) resulting from diabetes mellitus by changing NRF2/HO-1 signaling activity [115]. lncRNAs also have their own regulatory mechanisms in other eye diseases. Downregulation of lncRNA GAS5 resulted in increased expression of zeste homolog 2 and decreased expression of ATP-binding cassette transporter A1 in RGC cells, thereby inhibiting RGC cell apoptosis and alleviating the symptoms of glaucoma [116]. lncRNA Mbd2-AL1 targets miRNA-188-3p, thereby suppressing the expression of tumor necrosis factor receptor-associated factor 3 (Traf3) and attenuating ischemia/reperfusion injury-mediated apoptosis in retinal RGC cells [117].

### 4.4. Glial Cells

lncRNA XIST was downregulated in the retinal tissue of a DR model and could stabilize SIRT1, thereby inhibiting the activation of Müller cells and the production of inflammatory factors in DR [118]. The knockdown of lncRNA RNCR3 significantly inhibited the increase of retinal glial cells, decreased the proliferation activity of Müller cells, and reduced the expression of retinal glial response-related genes [119]. lncRNA MALAT1 promotes retinal microglia activation via the miR-124/MCP-1 signaling pathway and is involved in the inflammatory pathogenesis of DR [120]. Downregulation of lncRNA NEAT1 increased the concentration of miR-497, which resulted in decreased brain-derived neurotrophic factor expression, thereby promoting Müller cell apoptosis under HG conditions [121]. In addition to the significant effects of lncRNAs on glial cells in DR models, an emerging study has shown its role in other ocular diseases. lncRNA MALAT1 facilitates HG-induced inflammatory responses of microglial cells via the activation of MyD88/IRAK1/TRAF6 signaling [122]. lncRNA H19 initiates microglial pyroptosis by promoting miR-21/PDCD4 signaling in retinal ischemia/reperfusion injury [123]. lncRNA–Fendrr increased the pyroptosis of microglia by protecting NLRC4 from ubiquitination and degradation [124].

### 4.5. RPEs

lncRNA-XIST reduces apoptosis and restores the migration capability induced by HG by directly binding and inhibiting hsa-miR-21-5p expression in ARPE-19 cells [125]. lncRNA MEG3 inhibits RPE cell apoptosis and inflammation by suppressing miR-93 expression and regulating the miR-34a/SIRT1 axis [126,127]. lncRNA H19 reduces inflammation by inhibiting miR-93 expression via XBP1s in HG-induced RPE cells [128]. Overexpression of lncRNA LINC00673 can downregulate the expression of p53, thereby inhibiting the apoptosis of RPE cells in HG conditions [129]. lncRNA AK077216 reduces ARPE-19 cell apoptosis by inhibiting the expression of miR-383 [130].

### 4.6. Blood and Tears

lncRNA-OGRU expression is higher in the serum of patients with DR as compared with that in the serum of normal individuals. This phenomenon was confirmed by conducting experiments in rats with STZ-induced DR and HG-incubated Müller cells [131]. lncRNA-VIM-AS1 was remarkably decreased in the plasma of patients with type 2 diabetes mellitus compared with that in the plasma of healthy controls [132]. Compared with that in the controls, in peripheral blood samples of patients with DR, the expression of lncRNA SNHG16 and E2F1 were both increased and that of miR-20a-5p was decreased. lncRNA SNHG16 regulates E2F1 expression via the action of miR-20a-5p and aggravates PDR [133]. The content of lncRNA NR2F1-AS1 in the blood of patients with PDR was significantly increased. Blood lncRNA NR2F1-AS1 may play a role in regulating EndMT, and its expression can be a novel noninvasive biomarker for the early diagnosis of DR [134]. The expressions of lncRNA FLG-AS1 and miR-380-3p in the serum of patients with DR were negatively correlated, thereby regulating the expression of SOCS6. lncRNA FLG-AS1 overexpression attenuated retinal damage in an HG-treated rat model, modulated retinal epithelial cell inflammation, and reduced oxidative stress and apoptosis in ARPE-19 cells [135].

### 4.7. Exosomes

Exosomal lncRNA SNHG7 inhibits EndMT and tube formation in DR through the miR-34a-5p/XBP1 axis [136]. Therefore, supplementing exosomal lncRNA SNHG7 derived from mesenchymal stem cells, increasing its content in vivo, or using targeted therapy through the miR-34a-5p/XBP1 axis may be an effective means to treat DR.

## 5. Drug Regulation

ncRNA is a promising therapeutic target for DR and attracts lots of attention in drug development. An increasing number of studies have reported on such potential drugs.

### 5.1. Clinical Medicine

The high expression levels of Hsa-miR-3184-3p, hsa-miR-24-3p, and hsa-miR-197-3p were significantly reversed using aflibercept, an anti-VEGF drug [137]. 

### 5.2. Preclinical Medicine

The HuoXue JieDu formula is a classical formula that restores the retinal thickness and RGC number and decreases retinal cell apoptosis by regulating 15 miRNAs [138]; these miRNAs may be potential therapeutic targets for DR.

Isoliquiritigenin, a bioactive flavonoid, prevents the development and progression of DR by downregulating the expression level of retinal miR-195 [139]. Astragalus polysaccharide attenuates apoptosis in a concentration-dependent manner via the regulation of the miR-204/SIRT1 axis and miR-195 expression level in RPE cells [140,141]. Moreover, one study indicated that astragaloside IV protects RPE cells from apoptosis by upregulating miR-128 expression [142]. Baicalin protects against RPE cell injury caused by HG conditions by upregulating miR-145 expression [143]. Ginsenoside Rg1 can inhibit HG-induced mesenchymal activation and fibrosis by regulating miR-2113 signaling [144]. Another group reported that ginsenoside Rg1 protects against RPE cell apoptosis induced by HG conditions by upregulating miR-26a expression [145]. Resveratrol, a star molecule that is also a natural product, has attracted increasing attention worldwide. Zeng et al. revealed that resveratrol decreases diabetes-induced Müller cell apoptosis by upregulating miR-29b expression [146]. Melatonin suppresses Müller cell activation and proinflammatory cytokine production by upregulating the lncRNA MEG3-mediated miR-204/Sirt1 axis in experimental DR [147]. Danhong injection can upregulate miR-30d-5p and target JAK1 to ameliorate DR and kidney injury in HG-treated mouse models [148]. Hawthorn polyphenol extract reduces HG-induced apoptosis and inflammatory responses in ARPE-19 cells by modulating the miR-34a/SIRT1/p53 pathway [149]. Experimental data showed that treatment with Niaspan increases retinal miR-126 and Ang-1/Tie-2 levels in rats and can help repair retinal blood vessels, inhibit inflammation, repair blood–retinal barrier damage, and improve DR [150].

## 6. Perspective

DR is the main cause of vision loss in diabetic patients. The pathogenesis of DR involves pathological changes in various tissues, such as excessive cell proliferation and apoptosis, inflammatory responses, and oxidative stress. In recent years, studies have found that noncoding RNAs, including miRNAs, circRNAs, and lncRNAs, participate in the pathological process of DR through the abovementioned pathways. These noncoding RNA types can thus be used as biomarkers for the early detection and treatment of DR. ncRNAs can regulate the damage of retinal cells, such as ECs, pericytes, ganglion cells, glial cells, RPE cells, and the abnormal changes in AH, exosomes, extracellular vesicles, blood, tears, and VH, and neovascularization. Therefore, early diagnosis of DR can reduce its high incidence and ensure a good prognosis.

This article reviewed some ncRNAs that regulate DR progression and their mechanisms. Most circRNAs and lncRNAs negatively regulate their downstream miRNAs, which then regulate gene expression by regulating the transcription of the target genes. This affects the pathological processes of different cells, and a single ncRNA can regulate the expression of more than one target gene. Most of the aforementioned studies indicated that ncRNAs mainly bring about single-cell pathologic changes; this may help in understanding how ncRNAs mediate cell–cell crosstalk [85].

However, the roles of several circRNAs and lncRNAs in the treatment of DR have not yet been discovered, and their regulatory mechanisms remain to be explored. The current study mainly focuses on ECs, glial cells, RPE cells, etc. Studies on visual cells, bipolar cells, horizontal cells, amacrine cells, reticulum cells, etc., are limited. Second, DR is a chronic evolution process from early, middle, to late stages, and a model treated with HG may not be representative of all DR patients. Although new ideas for using ncRNAs in the treatment of DR are gradually being recognized, the transition from experimental research to clinical application remains a big challenge. Moreover, there is no unified naming standard for some ncRNAs; different researchers use different names to designate the same ncRNA during the research process, which is likely to cause confusion.

As described in this review, ncRNAs may be less toxic and more effective as targeted drugs than traditional therapeutic drugs. An in-depth study on the role of these molecules in DR may help identify novel biomarkers and provide future directions in the search for new therapeutic targets and new therapies.

## 7. Literature Search

A systematic literature search was performed using PubMed. The following keywords were searched: diabetic retinopathy, noncoding RNA, miRNA, circRNA, lncRNA, endothelial cell, pericyte, glial cell, retinal pigment epithelium cell, ganglion cell, aqueous humor, exosomes, extracellular vesicles, blood, tears, and VH. English abstracts or full articles were reviewed. Irrelevant articles were excluded. The references of all the included articles were screened for additional articles relevant to our topic.

## Figures and Tables

**Figure 1 biomolecules-12-01774-f001:**
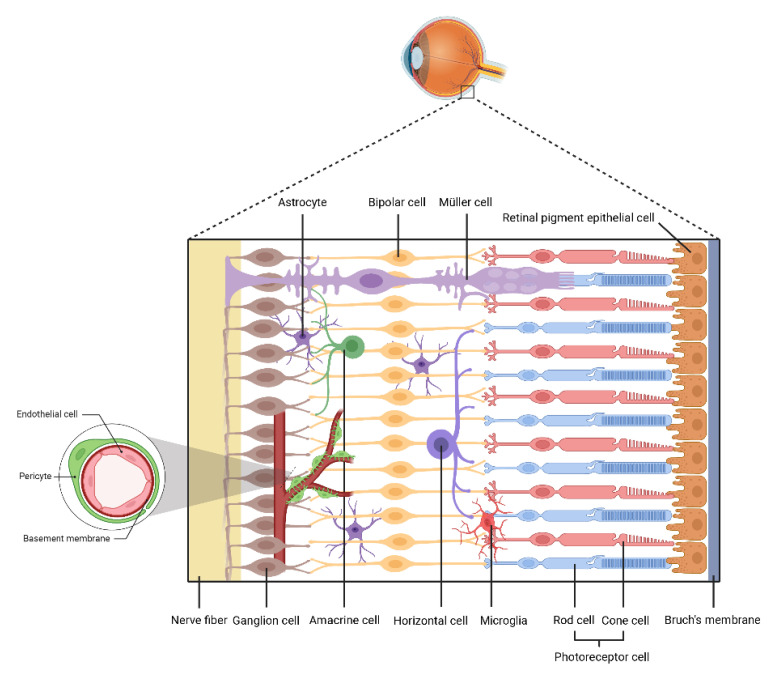
Primary structures in the retina. The retina comprises 55 different cell types, including pigment epithelium cell, photoreceptor cells, bipolar cells, ganglion cells, horizontal cells, amacrine cells, Müller cells, microglia, and astrocytes.

**Figure 2 biomolecules-12-01774-f002:**
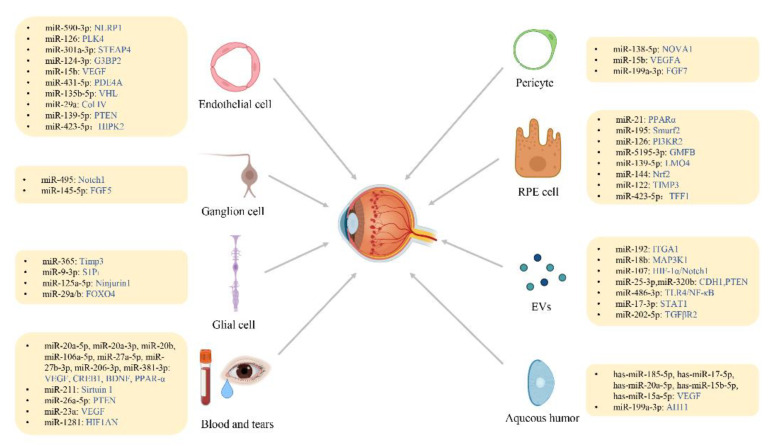
Schematic overview of miRNAs affecting DR-related cells and contributing to the pathogenesis of DR.

**Figure 3 biomolecules-12-01774-f003:**
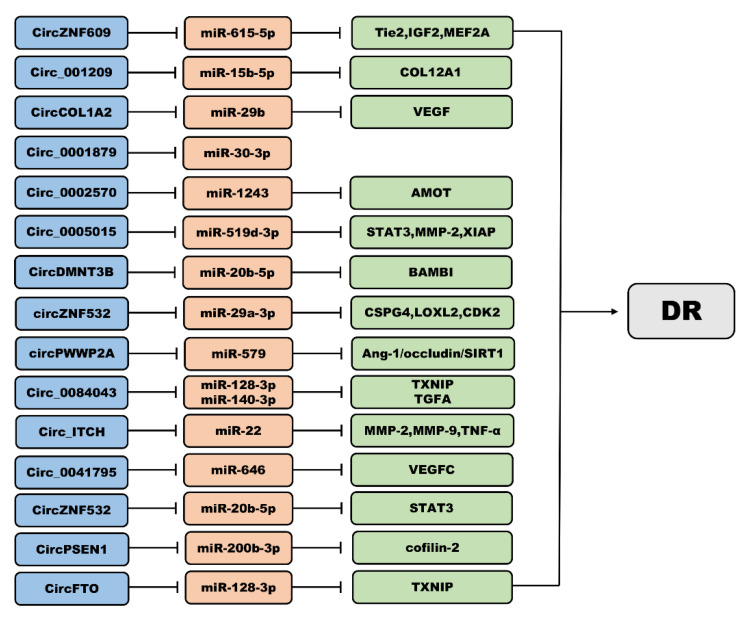
Main circRNAs involved in DR pathophysiology.

**Figure 4 biomolecules-12-01774-f004:**
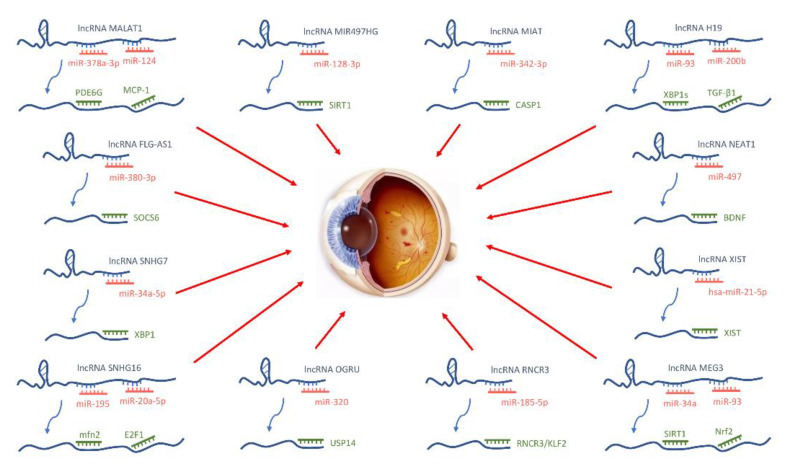
Schematic overview of lncRNAs promoting the pathogenesis of DR. IncRNAs regulate miRNAs in the progression of DR as well as the expression of related genes during transcription or epigenetic mechanisms to affect the progression of DR.

## Data Availability

Not applicable.

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
