# Peer review of "Noncoding RNAs Are Promising Therapeutic Targets for Diabetic Retinopathy: An Updated Review (2017–2022)"

_biomolecules, 2022, doi:10.3390/biom12121774_

Round 1

Reviewer 1 Report (Previous Reviewer 2)

The authors have addressed my prior concerns in a satisfactory manner. The review has been improved both in substance and style.

Author Response

Thank you for your comments.

Reviewer 2 Report (New Reviewer)

Basically, it is a well written review article to summarize the current stage of the noncoding RNAs' roles on diabetic retinopathy. However, the authors need to emphasize what has been updated in this article as there are lots of review papers regarding this work as far as I feel. It can be added in the introduction part before going to the main section.

Spell checking is needed; Horizontal cells (as far as I remember?) in Figure 1's illustration.

How about inflammatory cells? astrocytes? microglial cells in Figure 1's illustration? It needs to be added as the authors mentioned them in the manuscript.

English editing is recommended.

Author Response

Response to Reviewer 2 Comments

Dear reviewer:

Thanks for your letter and for your constructive comments concerning our manuscript. Those comments are all valuable and very helpful for revising and improving our paper. We have studied comments carefully and have made correction which we hope meet with approval. The main corrections in the paper and the responds to the comments are as follows:

Point 1: However, the authors need to emphasize what has been updated in this article as there are lots of review papers regarding this work as far as I feel. It can be added in the introduction part before going to the main section.

Response 1: Thank you for your suggestion. We have added relevant content. (Page 2, Line 62-67)

Point 2: Spell checking is needed; Horizontal cells (as far as I remember?) in Figure 1's illustration.

Response 2: Thank you for your suggestion. We have revised that. (Page 2, Fig.1)

Point 3: How about inflammatory cells? astrocytes? microglial cells in Figure 1's illustration? It needs to be added as the authors mentioned them in the manuscript.

Response 3: Thank you for your suggestion. We have modified Figure 1. (Page 2, Fig.1; Page 2, Line 48)

Point 4: English editing is recommended.

Response 4: Thank you for your suggestion. In this version, language use has been proofread by a native English speaker.

Thank you for your insightful comments. We hope that our responses and edits are satisfactory.

This manuscript is a resubmission of an earlier submission. The following is a list of the peer review reports and author responses from that submission.

Round 1

Reviewer 1 Report

The manuscript, “Noncoding RNAs are promising therapeutic targets for diabetic 2

retinopathy: an updated review (2017–2022)” summarizes a large number of publications based upon a literature search methodology. This is an important and interesting topic, but there are several problems with the manuscript as reviewed. Please find below a list of specific items.

Line 49 - laser treatment can slow vision loss in PDR and DME. The manuscript states NPDR and DME.

Line 73 - The sentence “miRNAs perform their functions by destabilizing and regulating the translation of their mRNA targets and protein, respectively, …” is confusing. miRNAs act by regulating the translation of their mRNA targets into protein. The way the sentence is written, it implies that miRNAs are interacting directly with the proteins, rather than indirectly by regulating the translation of the miRNA-targeted mRNAs.

Line 85 - STZ - you have to define acronyms when they are first used.

Line 102 - pericytes regulate communication between pericytes and ECs?

Line 103 - From Ref #25, pericytes are the focal point of DR pathogenesis. Not the focus point.

Line 107 - The terminology I am aware of is to call these retinal ganglion cells. I had to look up the term gangliocytes. It’s technically correct, but I would suggest changing it to retinal ganglion cells.

Line 117 - do you mean microglia instead of macroglia? The Muller cells, astrocytes, are both within the class of macroglia. So listing macroglia is redundant, and leaves out microglia.

Line 131 - The sentence one lines 130 - 131 implies that dysfunction of the RPE can lead to AMD followed by DR. This is not correct, and is never stated in Ref #40 which does not mention DR at all. I am not aware of literature showing that AMD precedes DR.  Once that is corrected, if the authors have a reference indicating RPE dysfunction leading to DR they should include that reference.

Given that exosomes are a subset of EVs, sections 2.7 and 2.8 should be combined.

There is a major problem with Reference #134, which is cited in reference to a formula that restored retinal thickness and RGC number. However, the reference included with the manuscript describes a staging strategy for COVID-19 infections.

Reference #43 is listed as an Invalid Citation in the References Section.

None of the figures appear to be referenced in the text. A description of the figure, and what it is conveying to the reader, should be included in body of the manuscript. In addition, more descriptive captions are needed for the figures.

In the description of Fig.1, the authors should note that the neovascularization shown in the lower portion of the figure represents abnormal anatomy, rather than a normal structure in the retina. Also, the retinal endothelial cells discussed in section 2.1 are not included in the figure.

In Fig.2 the label for Hyperlipidemia is mis-entered as Hhyperlipidemia.

There are two figures listed as Fig 4. One at line 263, and one at line 359.

Line 359 - A more descriptive caption is needed for Fig4, describing the inherent meaning behind the different structures (lncRNA vs miRNA vs genome and transcription). The re-use of the same red elements for miRNAs that are binding to a LNCRNA and for genes that are being transcribed will be confusing for the reader.

A point to consider - are these figures useful? They haven’t been referenced and there’s no clear location where why would be. 

Are there figures that could be included that would better serve the manuscript?

Reviewer 2 Report

The review by Mengchen Wang et al. attempts to summarize the considerable body of research into the involvement of non-coding RNAs in diabetic retinopathy. The goal of the authors to include all types of noncoding RNAs in the review, is quite ambitious, and makes it difficult to cover all relevant research, even if the review is restricted to the last 5 years. And indeed, many studies published in the field within this time frame, are omitted by the authors; for example, the described involvement in DR of miR-122 and miR-423. The authors should make sure to more systematically scan the relevant publications, and if necessary, narrow down the focus of the review. Additionally, language use should undergo proofreading by a native English speaker (for example, in the abstract, line 19, "contain" should be changed to "include" or "consist of". ) Finally, a minor comment - although the article is evaluated for Biomolecules, the template used throughout is that of a different MDPI journal.